# Repairing Annulus Fibrosus Fissures Using Methacrylated Gellan Gum Combined with Novel Silk

**DOI:** 10.3390/ma16083173

**Published:** 2023-04-18

**Authors:** Andreas S. Croft, Slavko Ćorluka, Janine Fuhrer, Michael Wöltje, Joana Silva-Correia, Joaquim M. Oliveira, Georg F. Erbach, Rui L. Reis, Benjamin Gantenbein

**Affiliations:** 1Tissue Engineering for Orthopaedics and Mechanobiology, Bone & Joint Program, Department for BioMedical Research (DBMR), Faculty of Medicine, University of Bern, CH-3008 Bern, Switzerland; andreas.croft@unibe.ch (A.S.C.); slavko.corluka@insel.ch (S.Ć.); janine.fuhrer@students.unibe.ch (J.F.); 2Department of Orthopaedic Surgery & Traumatology, Inselspital, Bern University Hospital, Medical Faculty, University of Bern, CH-3010 Bern, Switzerland; georg.erbach@insel.ch; 3Institute of Textile Machinery and High Performance Material Technology, Technische Universität Dresden, D-01062 Dresden, Germany; michael.woeltje@tu-dresden.de; 43B’s Research Group, I3Bs—Research Institute on Biomaterials, Biodegradables and Biomimetics, University of Minho, Headquarters of the European Institute of Excellence on Tissue Engineering and Regenerative Medicine, AvePark—Parque de Ciência e Tecnologia, Zona Industrial da Gandra, Barco, 4805-017 Guimarães, Portugal; joana.correia@i3bs.uminho.pt (J.S.-C.); miguel.oliveira@i3bs.uminho.pt (J.M.O.); rgreis@i3bs.uminho.pt (R.L.R.); 5ICVS/3B’s—PT Government Associate Laboratory, 4806-909 Braga/Guimarães, Portugal

**Keywords:** intervertebral disc herniation, methacrylated gellan gum, silk fibroin, annulus fibrosus repair, organ culture, complex dynamic loading

## Abstract

Intervertebral disc (IVD) herniation often causes severe pain and is frequently associated with the degeneration of the IVD. As the IVD degenerates, more fissures with increasing size appear within the outer region of the IVD, the annulus fibrosus (AF), favoring the initiation and progression of IVD herniation. For this reason, we propose an AF repair approach based on methacrylated gellan gum (GG-MA) and silk fibroin. Therefore, coccygeal bovine IVDs were injured using a biopsy puncher (⌀ 2 mm) and then repaired with 2% GG-MA as a filler material and sealed with an embroidered silk yarn fabric. Then, the IVDs were cultured for 14 days either without any load, static loading, or complex dynamic loading. After 14 days of culture, no significant differences were found between the damaged and repaired IVDs, except for a significant decrease in the IVDs’ relative height under dynamic loading. Based on our findings combined with the current literature that focuses on ex vivo AF repair approaches, we conclude that it is likely that the repair approach did not fail but rather insufficient harm was done to the IVD.

## 1. Introduction

Intervertebral disc (IVD) herniation is one of the most frequent diagnosed conditions associated with low back pain (LBP) [1]. Approximately 80% of the worldwide population will be affected by LBP at some point during their life, which consequently causes an enormous socioeconomic burden [2]. The incidence of IVD herniation itself ranges from 20 to 50 cases per 1000 adults annually, and its highest prevalence can be found among people aged 30–50 years [3,4]. IVD herniation is defined as an excessive displacement of the IVD’s highly hydrated core, the nucleus pulposus (NP), beyond the normal margins given by the surrounding tissue, the annulus fibrosus (AF) [5]. The most common causes of IVD herniation are IVD degeneration (IDD), followed by trauma [3]. On the one hand, IDD is part of the natural aging process [6,7]. On the other hand, genetic and environmental factors, excessive mechanical stress, as well as injury are also considered main drivers for the initiation and progression of IDD [8,9]. During IDD, the IVD shifts from predominantly anabolic reactions to an increased catabolic activity, resulting in a reduced cell number and decreased extracellular matrix (ECM) production [10]. As the IVD degenerates, more and more fissures with increasing size appear within the AF, which weaken it over time and create an ever-increasing challenge to keep the NP in place [11].

Interestingly, IVD herniations are frequently seen on MRIs of asymptomatic patients [3]. Nevertheless, people with hernia-related symptoms often complain about radicular pain, weakness, numbness, and sensation changes [3,5]. The induction of hernia-induced pain is believed to arise from mechanical compression on the nerve roots and/or from local inflammation induced by the protruding NP [12,13]. Although the AF itself only has a limited intrinsic healing potential, non-surgical treatments are sufficient in most cases to treat herniated IVDs and relieve the patient from LBP [14]. As usual, surgery is the last resort, and it is only applied in severe cases and in patients who fail conservative treatment [3]. Surgery comprises discectomy, which remains the gold standard for treating IVD herniation [15]. The problem with discectomy, however, is that it does nothing to reverse the process that made the herniation possible in the first place [16]. Consequently, the normal physiological structures of the IVD are not being restored, and there is a chance that the surgery further aggravates the existing damage in the AF [17]. Therefore, there is a need for novel AF repair approaches and strategies.

With its use of biomaterials, tissue engineering is emerging as one of the most promising approaches for the treatment of cartilaginous tissues, including the repair and regeneration of IVD tissue [18]. However, to fulfil these demanding requirements and to improve the chances of success, the combination of two or more biomaterials could be necessary. A promising biomaterial for tackling this challenge is silk fibroin, which has a relatively long and extensive history for IVD repair [19], as well as gellan gum (GG) and its derivatives. Silk fibroin has proven to be a very encouraging biomaterial because of its excellent biocompatible properties, low immunogenicity, and the fact that it can be processed into many different scaffold morphologies [20,21]. For the regeneration of the NP, silk fibroin has mainly been used as a hydrogel, whereas firm scaffolds have been the preferred choice for regenerating the AF in past studies [19]. GG, the second biomaterial used in this study, is an exopolysaccharide which is produced through microbial fermentation by bacteria from the *Sphingomonas* group [22]. What makes GG so interesting is that it exhibits many properties that are very favorable as a biomaterial for tissue engineering, including great biocompatibility, high hydration, a low production cost, thermo-reversibility, injectability, and the option to chemically modify it to change its functionality [18,23]. One of the most commonly performed chemical alterations on GG is methacrylation [23]. Methacrylated GG (GG-MA) has the great benefit of enhancing the gel’s mechanical performance and improving its manageability at physiological temperatures, therefore making it relevant for IVD-related tissue engineering applications [24,25]. Previous studies have shown promising results using GG-MA to regenerate the IVD’s NP [25,26]. However, it is unknown how suitable the biomaterial is for repairing and/or regenerating the AF.

For these reasons, the aim of this bovine IVD organ culture study was to come up with an approach to repairing damaged AF tissue using 2% GG-MA as a filler material and sealing it with an embroidered silk yarn fabric, which resembles the macroscopical structure of natural AF tissue. The damage applied to the IVDs should replicate the fissures found in herniated IVDs. Damaged, repaired, and untreated control IVDs were cultured for 14 days either without any load, static loading, or complex dynamic loading to mimic physiological loads.

## 2. Materials and Methods

### 2.1. Dissection of Bovine IVDs

Caudal IVDs were dissected as previously described by Chan et al. [27]. Here, the IVDs were extracted from the tails of approximately one-year-old cows, which had been sacrificed on the same day. Therefore, the bovine tail was first immersed into 1% Betadine^®^ solution (Mundipharma Medical Company, Basel, Switzerland) for superficial disinfection. In the next step, the tissue surrounding the IVDs was removed using a sterile scalpel blade, and the exposed IVDs were subsequently dissected from the tail using a custom-made guillotine. Once at least seven IVDs were dissected, a caliper was used to measure the IVDs’ diameter at two points 90° apart, and their height was measured from one CEP to the other CEP. Then, all IVDs were rinsed with Ringer’s lactate solution (#FE1010206; Bichsel, Interlaken, Switzerland) using a Zimmer Pulsavac PlusTM jet-lavage system (Zimmer Biomet, Inc., Winterthur, Switzerland). To sterilize the IVDs, each sample was immersed twice in penicillin/streptomycin (1000 U/mL; #5711; Sigma-Aldrich, Buchs, Switzerland) for five minutes and then washed with phosphate-buffered salt solution (PBS). Finally, all IVDs were placed into a sterile beaker containing high-glucose (4.5 g/L) Dulbecco’s Modified Eagle Medium (HG-DMEM; #52100-039; Gibco, Life Technologies, Zug, Switzerland), supplemented with 5% fetal bovine serum (FBS; #F7524; Sigma-Aldrich), 0.22% sodium hydrogen carbonate (#31437-500G-R; Sigma-Aldrich), 10 mM HEPES buffer solution (#15630-056; Thermo Fisher Scientific, Basel, Switzerland), 1 mM sodium pyruvate (#11360-039; Thermo Fisher Scientific), and penicillin/streptomycin/glutamine (100 U/mL, 100 μg/mL, and 292 μg/mL, respectively; #10378-016; Thermo Fisher Scientific). In this state, the samples were incubated at 37 °C with 20% O_2_ and 5% CO_2_ for one day.

### 2.2. Assembly of the Embroidered Silk Yarn Patches

Twisted schappe silk yarn of a fineness of 140/2 Nm (number metric) (Plauener Seidenweberei GmbH, Plauen, Germany) was used for braiding silk yarn patches. The patches were assembled on a ZSK Racer 1W embroidery machine (ZSK, Stickmaschinen GmbH, Krefeld, Germany). The embroidered patches were each composed of five layers of silk yarn. The distance of the yarn deposit was set at 0.7 mm, and the silk yarn in each layer was embroidered with alternating orientations of −30° and +30° on the water soluble embroidery ground “Solvy Fabric No. 41825” (company: Gunold GmbH, Stockstadt, Germany) according to the natural arrangement [28]. For both the upper and the lower yarn, the same silk yarn was used. These two yarn systems together formed the layer structure for the individual layers, which together build up the annulus fibrosus patch. After embroidery, the silk patches were separated from the embroidery ground by a two-step washing procedure. In the first step, three washing cycles were carried out. For this purpose, the specimens were soaked in lukewarm water in a washbasin for about 5 min each and then placed in a beaker filled with water. This procedure was repeated three times. The last rinse was carried out with distilled water in the beaker. In the second step, the pre-dissolved samples were placed in a water-permeable wire basket, which was placed in a beaker and stirred for three days at 50 °C and 300 rpm in distilled water. Finally, the water was changed again, and the patches were dried for one day at 60 °C.

### 2.3. Production of Methacrylated Gellan Gum

GG-MA was prepared according to a previously described protocol [25] by reacting low-acyl gellan gum (LAGG; Gelzan^TM^ CM Gelrite^®^, Sigma-Aldrich) with glycidyl methacrylate (GMA; 97%, Sigma-Aldrich). Briefly, GG was completely and homogeneously dissolved in ultrapure water at 90 °C under constant agitation at a final concentration of 1% (*w*/*v*). After the complete dissolution of LAGG and the cooling of the solution to room temperature, an appropriate amount of GMA was added. The reaction occurred for the period of 24 h at room temperature under vigorous stirring, after adjusting the pH to 8.5 (periodic adjustments were also performed during the reaction). Afterwards, 1/2 volume of cold acetone was used to precipitate the resulting reaction products, which were then purified by dialysis (cellulose membrane, molecular weight cut-off 12 kDa, Sigma-Aldrich) against distilled water for four days. The purified GG-MA was frozen at −80 °C and freeze-died for a minimum of seven days. The obtained GG-MA powders were sterilized under an ethylene oxide gas atmosphere and stored, protected from light, in a dry place until further use.

### 2.4. AF Damage and Repair

After one day of incubation, the IVDs were assigned either to the: (i) control, (ii) damage, or (iii) repair groups. The IVDs in the control group were left untreated. However, the IVDs in the other two groups were punctured with a single 2 mm circular biopsy punch (#BPP-20F; Kai Industries Co., Ltd., Seki City, Japan, distributed by Polymed Medical Center, Glattbrugg, Switzerland) to replicate an AF fissure [29]. To repair the IVDs, GG-MA was pre-heated to 60 °C and then injected into the cavity. Next, the GG-MA had to be cross-linked, and the defect had to be sealed (Figure 1). Therefore, one of the silk fibroin AF patches was soaked into PBS, placed onto the defect, and sutured onto the IVD, as previously described by Li et al. [30]. Cross-linking of the hydrogel could be achieved through the interaction between the PBS in the AF patch and the GG-MA itself.

### 2.5. Organ Culture

As soon as all the IVDs were either damaged and/or repaired or untreated (control), they were assigned to one of three different mechanical loading regimes, i.e., (i) no load, (ii) static loading, (iii) complex dynamic loading, and culturing for 14 days. IVDs that experienced static loading were exposed to a persistent pressure of 0.2 MPa. However, IVDs that underwent the complex dynamic loading regime were exposed to a “low-pressure” loading regime, consisting of an active phase for 8 h and a passive phase for 16 h (Figure 2). This cycle was repeated every 24 h and was intended to mimic the diurnal mechanical loading profile that IVDs are physiologically exposed to in the spine. During the active phase, the IVDs were subjected to a sinusoidal pressure (0.2 MPA ± 0.1 MPa at 0.1 Hz) combined with torsion (±2° at 0.2 Hz). During the passive phase, however, the IVDs were allowed to recover, as they were subjected to a reduced static pressure of 0.1 MPa without torsion. The dynamic loading regime was executed using a custom-built bioreactor that enables two degrees-of-freedom movement.

### 2.6. Tissue Activity and Digestion

After two weeks of organ culture, one of the CEPs was removed to gain better access to the IVD’s AF. Here, the outer AF was divided into quarters, two of which were adjacent to the damage/repair made to the AF (damaged side) and two quarters were on the opposite side (intact side). One fragment from each side was collected for RNA extraction, and the remaining two fragments were collected to measure the tissue’s metabolic activity. To determine to tissue’s metabolic activity, the samples were immersed in 500 µL of 50 µM resazurin sodium salt solution (#R7017-1G, Sigma-Aldrich) and incubated for 1.5 h at 37 °C [32]. After the incubation time, the samples were transferred into PBS, and the fluorescence of the resazurin sodium salt solution was read at an excitation wavelength of 544 nm and an emission wavelength of 578 nm using an ELISA reader (Spectramax M5, Molecular Devices, distributed by Bucher Biotec, Basel, Switzerland). Finally, outcomes were normalized to the tissue’s dry weight and DNA content.

As soon as the metabolic activity was determined and the AF samples were washed with PBS, they were dried overnight at 60 °C. The following day, the dry weight of each sample was identified, and then they were digested overnight at 60 °C in a papain solution (3.9 U/mL; #P3125; Sigma-Aldrich) supplemented with 5 mM L-cysteine hydrochloride (#20119; Sigma-Aldrich) [29].

### 2.7. Glycosaminglycan (GAG) Content

The amount of sulphated glycosaminoglycan (GAG) was determined in the digested AF tissue as well as in the culture medium, which was collected at day 13, three days after the last medium change. Therefore, 1,9-dimethyl-methylene blue zinc chloride double salt dye (#341088; Sigma-Aldrich) was added to each sample and to the standard, which was based on chondroitin sulfate sodium salt from bovine cartilage (#6737, Sigma-Aldrich) [33]. The absorbance was measured within five minutes at a wavelength of 600 nm. Once the amount of GAG was determined, the content in the culture medium was normalized to the IVD’s initial volume after its dissection, and the tissue’s GAG was normalized to its dry weight and DNA content.

### 2.8. DNA Content

To quantify the amount of DNA in the digested tissue samples, Hoechst 33258 dye (#86d1405; Sigma-Aldrich) was applied to all samples and to the standard, which was made of DNA sodium salt from calf thymus (#D1501; Sigma-Aldrich). The fluorescence was measured at a wavelength of 350 nm excitation and 450 nm emission [34].

### 2.9. Nictric Oxide Content

The number of nitric oxide (NO) radicals produced and released by each IVD was measured from the same culture medium, which has also been used to determine the GAG content. Since NO is very short-lived, its content was indirectly quantified by measuring its oxidized product nitrite (NO_2_^−^) using the Griess reaction [35]. Therefore, proteins in the culture medium were precipitated with absolute ethanol, and its supernatant was mixed with N-(1-naphthyl)ethylenediamine dihydrochloride (#N9125; Sigma-Aldrich), sulfanilamide (#S9251; Sigma-Aldrich), and 10% phosphoric acid (#79617; Sigma-Aldrich). A standard was made using NaNO_2_, and the absorbance was then measured at a wavelength of 530 nm. To exclude any kind of unspecific absorbance, the samples were further measured at a wavelength of 650 nm. Finally, the results were normalized to the IVDs’ volume after their dissection from the tail.

### 2.10. RNA Extraction and qPCR

The RNA was extracted using a modified TRIspin method [36]. Therefore, snap-frozen AF tissue samples from the damaged and intact side of each IVD were first ground to a powder using a mortar and pestle. The pulverized tissue was then added into TRIzol reagent^®^ (#TR118; Molecular Research Center; Cincinnati, OH, USA, distributed by Lucerna-Chem Inc., Lucerne, Switzerland), which was enriched with a polyacryl carrier (#PC152; Molecular Research Center). Next, phase separation was induced using 1-Brome-3-Chloropropane (BCP; #B9673; Sigma-Aldrich), from which the clear phase was mixed with molecular grade absolute ethanol (#51976; Sigma-Aldrich). The RNA was then extracted using a GenElute miniprep kit (#RTN70; Sigma-Aldrich), and genomic DNA was digested with an On-Column DNase I Digestion Set (#DNASE70; Sigma-Aldrich). Afterwards, reverse transcription from RNA to cDNA was achieved using a High-Capacity cDNA Reverse Transcription kit (#4368814; Thermo Fisher Scientific) and a MyCycler™ Thermal Cycler system (#1709703; Bio-Rad Laboratories; Cressier, Switzerland). Then, quantitative polymerase chain reaction (qPCR) was carried out with iTaq Univeral SYBR Green Supermix (#1725122; Bio-Rad) with the primers of interest (Table 1) using the CFX96™ Real-Time System (#185-5096; Bio-Rad Laboratories). Finally, the relative expression was determined using the 2^−ΔΔCt^-method [37], and ribosomal 18S was used as a reference gene. Here, the no-load, static loading, and complex dynamic loading regimes were normalized to the untreated control IVDs, which were set to a relative gene expression of 1.

### 2.11. Statistics

A nonparametric distribution was assumed for all quantitative data. Usually, the data are shown as the mean ± standard deviation (SD). Exceptions are the qPCR data, which are presented as the mean + SD, and the absolute IVD heights are shown with their mean only. A Kruskal–Wallis test followed by Dunn’s multiple comparisons test was applied to all data except for the absolute IVD height, where data were analyzed by Repeated Measures (RM) two-way ANOVA followed by Sidak’s multiple comparisons test. All the statistical tests were carried out using GraphPad Prism (version 9.4.1 for Mac OS X, GraphPad Software; San Diego, CA, USA), and a *p*-value < 0.05 was considered statistically significant. Up to seven replicates were used for each experiment; however, the exact number of biological replicates (*n*) is indicated in the figure legends.

## 3. Results

### 3.1. Changes in the IVD Height

After 14 days of culture, all IVDs showed significant changes in their absolute height, regardless of whether the IVDs were repaired or not (Figure 3a). However, the IVDs that were cultured without any additional mechanical loading significantly increased their height (control: *p* < 0.05, “no load damage”: *p* < 0.01, and “no load repair”: *p* < 0.05), whereas the IVDs under mechanical loading significantly decreased their height (*p* < 0.0001). Since any significant positive or negative change from the initial IVD height can be considered pathological, the IVDs’ height changes were calculated relative to their initial height after dissection from the bovine tail. After 14 days, similar outcomes could be observed. Compared to the controls, a significant (*p* < 0.05) height loss was found with damaged IVDs under dynamic loading, and a strong trend in height loss (*p* = 0.097) was seen with damaged IVDs under static loading (Figure 3b). Finally, although not significant, repaired IVDs under mechanical loading generally showed an improved height recovery compared to damaged samples (damaged up to ø 19.9% and repaired up to ø 17.9% height decrease), and repaired IVDs without any loading appeared to have more similar height changes to the controls than damaged IVDs (control: ø 4.7%, “no load damage”: 6.3%, and “no load repair”: 5.1% height increase, respectively).

### 3.2. Metabolic Activity

To determine the influence of the repair approach on the tissue’s metabolic activity, it was compared to the tissue of damaged IVDs and untreated control IVDs. Here, the metabolic activity remained stable and showed comparable results throughout all conditions (Figure 4). Neither the damage or repair approach nor the mechanical loading regime significantly influenced the metabolic turnover. Moreover, the tissue’s metabolic activity displayed a very similar trend regardless of whether the activity was normalized to its dry weight (Figure 4a) or to its DNA content (Figure 4b).

### 3.3. Glycosaminoglycan Content

The GAG content was determined in the AF tissue and in the culture medium. However, no significant differences were found between the different conditions, as all samples contained a similar amount of GAG as the controls (Figure 5). Only the amount of GAG released into the culture medium from the repaired IVDs without any mechanical loading showed a slight but non-significant decrease (*p* = 0.31, Figure 5c). Furthermore, and similar to the metabolic activity, the outcomes for the GAG content in the tissue displayed the same trend, regardless of whether the activity was normalized to the tissue’s dry weight (Figure 5a) or to its DNA content (Figure 5b).

### 3.4. Nitric Oxide Content

The amount of NO produced by the IVDs and released into the culture medium did not significantly differ between the conditions (Figure 6). Nevertheless, damaged samples tended to produce somewhat more NO than their repaired counterparts.

### 3.5. Gene Expression

A qPCR was performed to determine the influence of the damage and repair approach under the impact of different mechanical loading regimes on the AF tissue’s relative gene expression. Therefore, anabolic mediator genes (*ACAN*, *COL1*, and *COL2*) (Figure 7a), catabolic mediator genes (*ADAMTS4*, *MMP3*, and *MMP13*) (Figure 7b), inflammation markers (*COX2*, *IL-1β*, and *RANTES*) (Figure 7c), and genes sensitive to mechanical stimuli (*COMP*, *CILP*) were analyzed (Figure 7d).

Concerning the anabolic markers tested, repaired IVDs generally showed a higher but not significant upregulation of *ACAN* and *COL2* compared to damaged samples (Figure 7a). The biggest differences were seen with *COL2* under dynamic loading on the intact side of the IVD (“dynamic load damage”: 5.6 ± 6.4-fold upregulation vs. “dynamic load repair”: 19.0 ± 29.7-fold upregulation). No differences at all were observed between tissue from the damaged and intact side as well as between the different loading regimes. Catabolic mediator genes remained largely unchanged; at most, under the mechanical load, they were marginally upregulated (Figure 7b). Inflammatory genes were generally, although not significantly, upregulated regardless of the condition (Figure 7c). Finally, a minor trend was observed with the mechanosensitive marker *COMP*, which was only upregulated when dynamic mechanical loading was applied on the IVDs (“dynamic load damage”: up to 13.9 ± 21.9-fold upregulation and “dynamic load repair”: up to 9.0 ± 11.1-fold upregulation) (Figure 7d).

## 4. Discussion

In this study, we managed to come up with an approach to repairing damaged AF tissue using 2% GG-MA as a filler material and sealing it with an embroidered silk yarn fabric. During the culture period of 14 days, the silk patch always stayed in place and managed to withstand the static and complex dynamic forces that were applied to the IVD. Interestingly, it seems like the repair approach had its greatest impact on the IVDs’ height, since the only significant differences in this study were found here. Although all IVDs under mechanical loading displayed a height loss beyond what is considered to be physiological (~10% height loss) [30,38], only the damaged IVDs under dynamic loading showed a significant decrease in relative disc height, whereas the repaired counterpart did not significantly differ from the control. Therefore, we can conclude that the repair approach positively influenced the retention of the initial disc height. In terms of cell activity, GAG content, NO production, or the gene expression profile, all downstream analyses lead to the conclusion that the repair approach did not significantly influence the IVDs’ general state compared to the damaged samples. Furthermore, injuries did not affect the IVD locally as no significant differences were detected between tissue samples from the damaged side and the contralateral intact side.

The biopsy punch size of 2 mm was chosen based on a study by Frauchiger et al. [29], which compared the effect of two AF-damage models, one of which being the 2 mm biopsy punch model. It was considered as a “very severe injury”, too severe to promote an inflammatory or repair response. In a follow-up study, it was then investigated how such a “severe” damage could be repaired ex vivo [34]. Like in our study, caudal bovine IVDs were damaged with a 2 mm biopsy punch and cultured for 14 days under no load, static loading, and complex dynamic loading. Indeed, this 14-day ex-vivo culture period, and also most of the mechanical loading parameters for the static and complex dynamic loading regimes, served as a foundation for our study. Although, in contrast to our study, their dynamically loaded IVDs were subjected to a “free swelling” period for the majority of the culture period, a state that naturally cannot be found in caudal/lumbar bovine and human IVDs in situ [39,40]. Furthermore, they used genipin-enhanced fibrin to fill the cavity and then sealed the site with a silk fleece-membrane, however, without a tissue-like structure. Their outcomes revealed that the IVD height could not be recovered, GAG contents did not significantly differ between damaged, repaired, and untreated control discs, and comparable gene expression levels were detected on the injured and intact sides of each IVD. Moreover, the genipin turned out to be cytotoxic in in vitro tests.

For this reason, our study tried to build on and improve upon the foundation made by Frauchiger et al. As a result, we exchanged the cytotoxic genipin with GG-MA, which is known for its great biocompatibility [24], and replaced the unspecific silk-fleece with an embroidered silk yarn patch, which was tailor-made to morphologically mimic the AF. Although some achievements were made with the IVDs’ height retention and AF patch’s ability to stay in place, other outcomes were not significantly different from the damaged IVDs or the control samples and were very much in line with what Frauchiger et al. found. Hence, at first glance, it appears that the repair approaches did not significantly improve the IVDs’ general state. Upon closer inspection, however, it becomes more obvious that it is likely that it is not the repair approach that failed but rather that insufficient harm was done to the IVDs. Consequently, the damaged IVDs resembled too much of their control condition, and thus, there was no significant damage that needed to be repaired. Therefore, we suggest that the AF damage was not “too severe”, as stated by Frauchiger et al., but exactly the opposite, namely, too mellow.

A study that investigated a larger cavity in the AF was conducted by Li et al. [30]. In that study, the AF was damaged with a 4 mm biopsy puncher, the cavity was then filled with a hyaluronan-based hydrogel, and the defect was closed with the outer AF tissue that had been punched out. After five culture days, the repair showed limited success, as a significant height loss, a significant decrease in compressive stiffness, and a lower cell viability compared to the controls were observed. Nevertheless, the IVD height managed to recover after the free swelling period. In contrast to our study, however, the mechanical loading regimes were shorter (six hours per day and five days in total) and more moderate (0.06 MPa ± 0.02 at a frequency of 0.1 Hz), and the IVDs were subjected to no load at all for the majority of the culture period.

Interestingly, even though ex vivo studies using IVD explants have had little impact on the IVDs’ healing process, recent long-term in vivo models show very promising results. For example, genipin-enhanced fibrin was able to prevent the IVD from herniating, maintained the IVD’s height, integrated with the surrounding AF tissue, and induced fibrous healing twelve months after it was injected into a damaged ovine IVD [41]. Another recent in vivo study used biodegradable scaffolds made of polyglycolic acid and hyaluronic acid to repair a partially resected ovine IVD [42]. Compared to the untreated controls, the implant was responsible for more intense Safranin-O staining and significantly more lamellae tissue structure three months post-surgery. However, what both in vivo studies have in common is their relatively long duration of multiple months, especially compared to the one to two weeks given in the ex vivo studies. Therefore, on the one hand, it is difficult to predict how the explanted IVDs would have behaved if a three-month culture period had been possible. On the other hand, it is also unknown how far the healing process in the in vivo models had already progressed two weeks post-surgery. Consequently, a fair direct comparison between the ex vivo and in vivo studies is difficult.

Although the application of GG-MA did not lead to the desired outcomes in this study, GG has generally been considered as a great candidate for the repair and regeneration of cartilage and IVD tissue [25,43]. Oliveira et al. used GG to encapsulate human nasal chondrocytes [44]. After a culture period of two weeks, the viscoelastic hydrogel was found to be non-cytotoxic, allowed chondrocytes to form clusters, and facilitated the synthesis of ECM. A follow-up study then tested the in vivo properties of GG [45]. In that study, GG alone or as a carrier for autologous adipose-derived stromal cells or articular chondrocytes was injected into rabbits with full-thickness articular cartilage defects and left for eight weeks. Histological evaluation showed that the hydrogel managed to integrate with the adjacent tissue under all conditions. The weakest integration, however, was found with acellular GG. Notably, although the cell-laden hydrogels performed, unsurprisingly, better than the empty hydrogels in terms of de novo tissue formation and its quality, acellular GG still displayed some focal spots of hyaline-like chondrocytes and was also able to induce the formation of newly formed tissue. This indicates that neighboring cells successfully managed to migrate from the tissue into the hydrogel.

The mentioned studies by Oliveira et al. directly address one of the limitations we faced in this study. All our repair conditions were carried out with acellular GG-MA. Probably, the best approach would have been to encapsulate autologous mesenchymal stromal cells (MSCs) into the hydrogel. However, as we only had access to the bovine tail, this was not an option. Even if MSCs could be extracted from the same donor from which the IVDs were dissected, it is likely that the cells would have had to be expanded first to obtain a sufficient cell number for injection. This, however, would not have been possible, as the IVDs were already treated the day after their dissection. A second limitation of this study is a lack of downstream analysis of the hydrogel itself. This was mainly due to the elusive nature of hydrogel but also due to the small defect size and consequently small amount of GG-MA that we could inject into the defect. Moreover, it would have been questionable if enough RNA could have been extracted from cells that potentially migrated into the hydrogel. For these reasons, we decided to analyze the tissue directly next to the damage/repair site as well as the tissue of the contralateral intact side.

## 5. Conclusions

In this study, we came up with an AF fissure repair approach using 2% GG-MA as a filler material and an embroidered silk yarn fabric to seal the filled cavity. GG-MA has been proven to be a suitable biomaterial for injection into damaged AF tissue, and the hydrogel could be cross-linked at the injection site. Furthermore, the silk patch always stayed in place, regardless of the loading regime. Nevertheless, this repair approach did not manage to significantly improve the general state of the IVDs compared to damaged samples. On the one hand, this seems to be a general problem with ex vivo AF repair approaches, as many research groups struggle to find a solution to restore this tissue. On the other hand, in vivo studies have recently shown very promising results, with evidence of tissue regeneration. Hence, in the future, we need to either focus on in vivo strategies for regenerating the AF or reconsider how the IVDs can be better cultured ex vivo. The better we can mimic the IVDs’ natural environment, the longer the IVDs can be cultured for ex vivo and the more successful we may become in finding a suitable AF repair approach.

## Figures and Tables

**Figure 1 materials-16-03173-f001:**
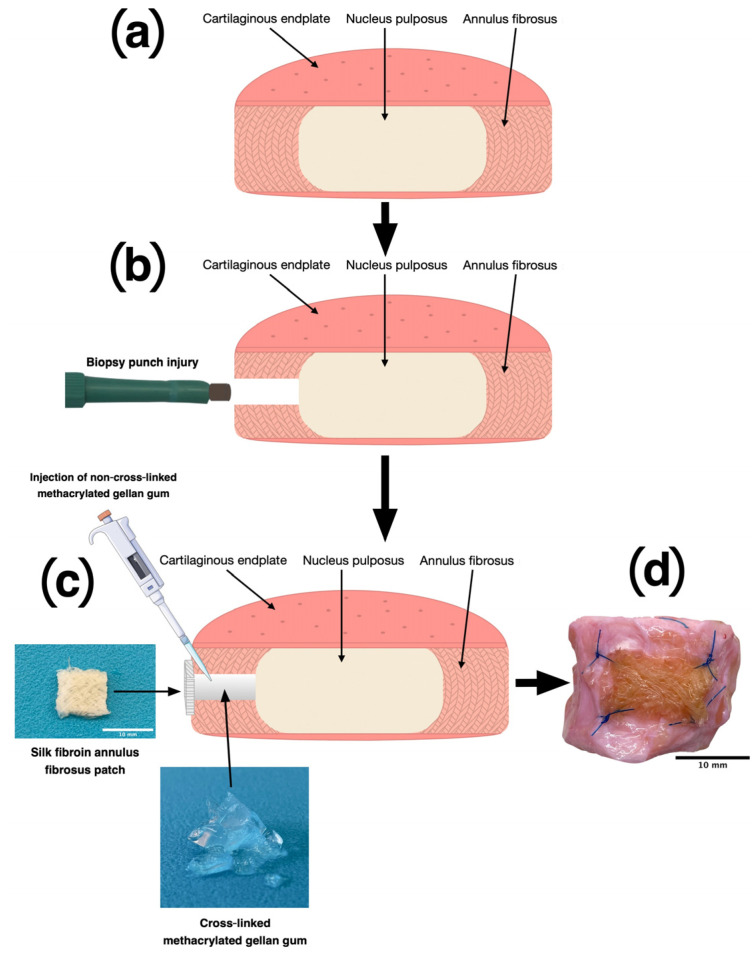
(**a**) Cross-section of a healthy bovine intervertebral disc with nucleus pulposus, annulus fibrosus, and cartilaginous endplates, (**b**) damage of the annulus fibrosus using a biopsy puncher, (**c**) repair of the damage using 2% GG-MA and a silk yarn fabric, (**d**) photograph of a repaired intervertebral disc. Adapted with permission from [31], © MDPI, Basel, IJMS.

**Figure 2 materials-16-03173-f002:**
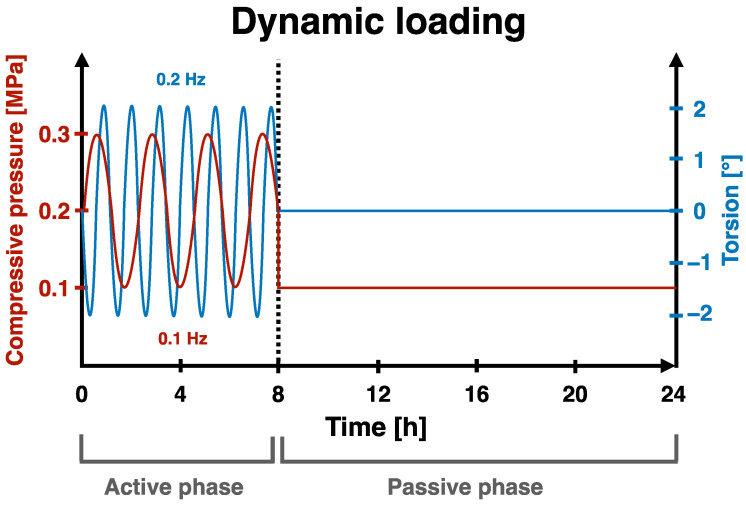
Representation of the daily dynamic loading regime, which was repeated for 14 days. The red line represents the compressive pressure, and the blue line represents the torsion.

**Figure 3 materials-16-03173-f003:**
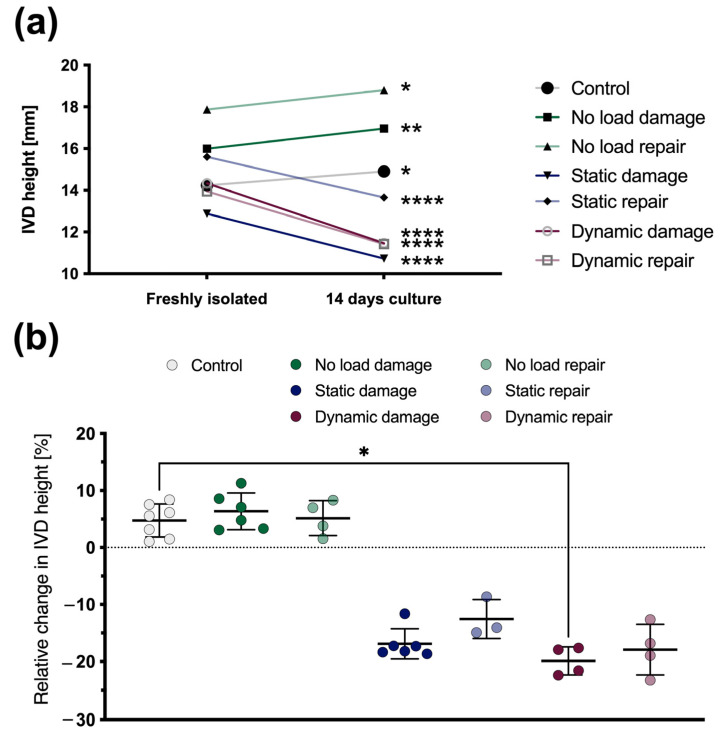
(**a**) Absolute IVD height changes either without additional mechanical loading, under static loading, or under complex dynamic loading for 14 days. Mean, *n* = 3–7. (**b**) Relative IVD height changes either without additional mechanical loading, under static loading, or under complex dynamic loading for 14 days. Mean ± SD, *n* = 3–7, *p*-value: * < 0.05, ** < 0.01, **** < 0.0001.

**Figure 4 materials-16-03173-f004:**
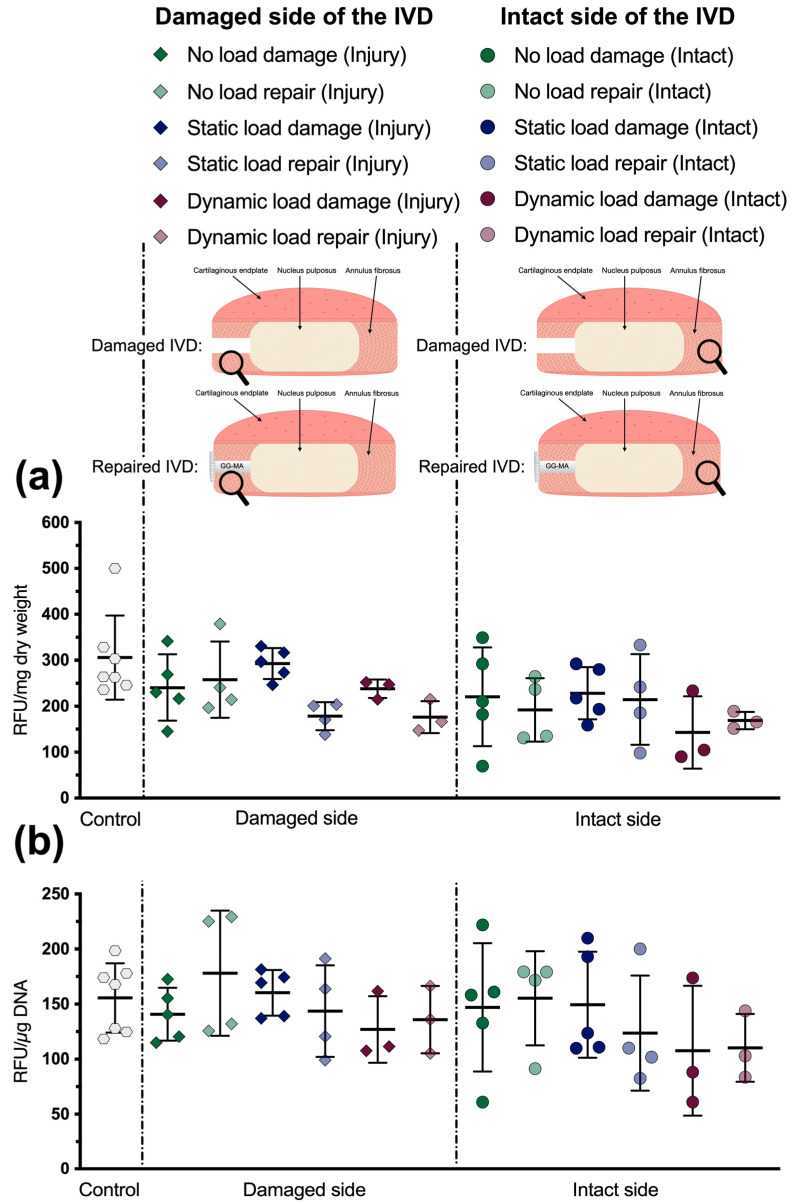
(**a**) Metabolic activity of AF tissue samples relative to their dry weight. (**b**) Metabolic activity of AF tissue samples relative to their DNA content. Mean ± SD, *n* = 3–7. Adapted with permission from [31], ©2021, MDPI Basel, IJMS.

**Figure 5 materials-16-03173-f005:**
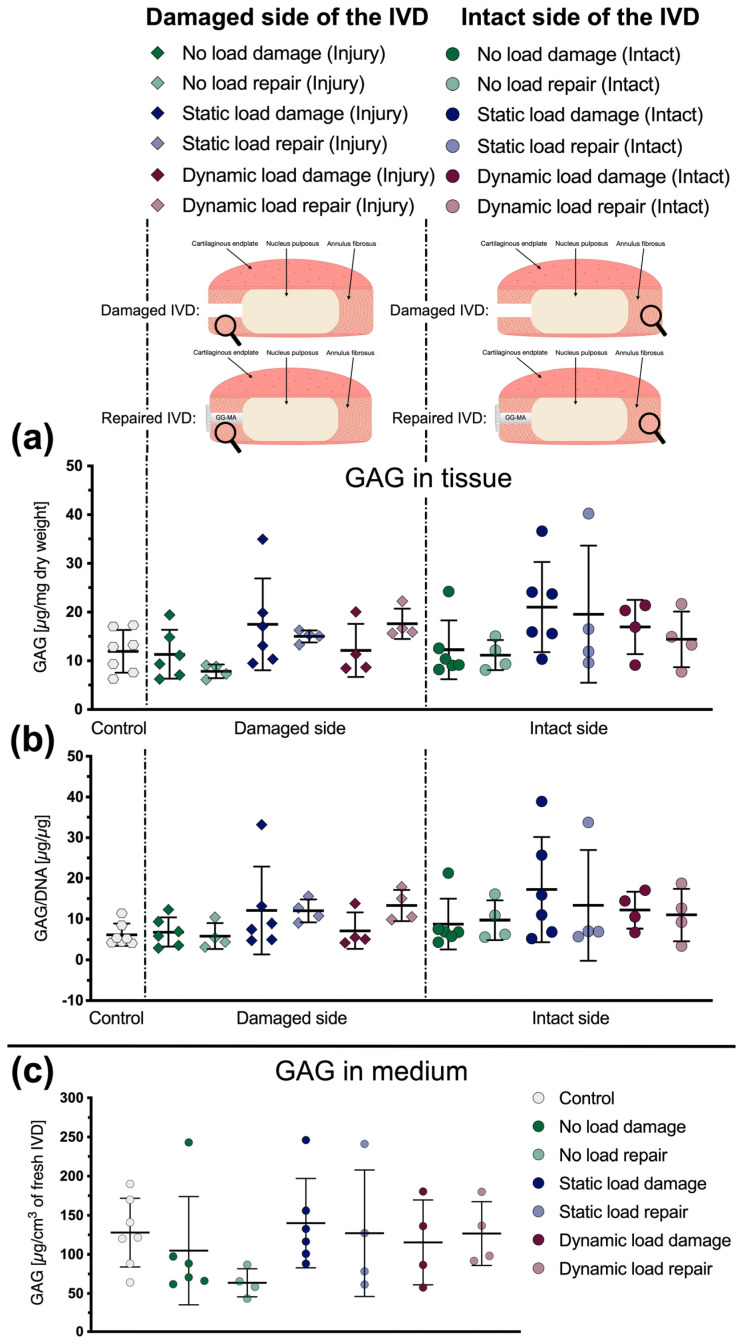
(**a**) GAG content of AF tissue samples relative to their dry weight. (**b**) GAG content of AF tissue samples relative to their DNA content. (**c**) Amount of GAG released from the IVDs into the culture medium relative to the IVDs’ volume after dissection from a bovine tail. Mean ± SD, *n* = 4–7. Adapted with permission from [31], © 2021, MDPI Basel, IJMS.

**Figure 6 materials-16-03173-f006:**
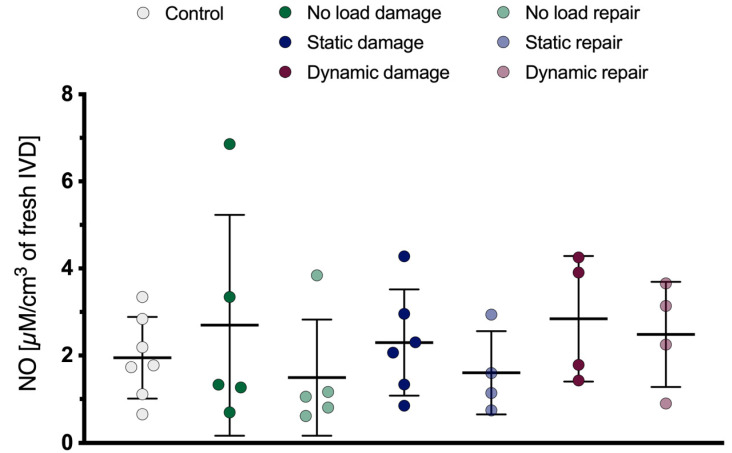
Amount of NO released from the IVDs into the culture medium relative to the IVDs’ volume after dissection from a bovine tail. Mean ± SD, *n* = 4–7.

**Figure 7 materials-16-03173-f007:**
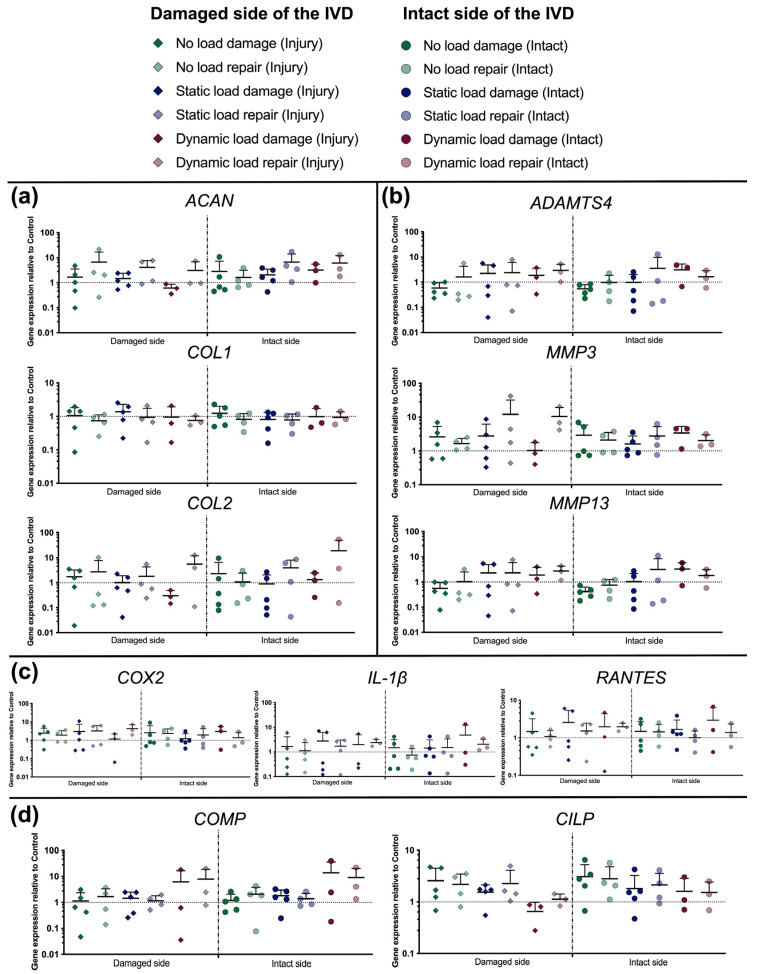
Relative gene expression of: (**a**) anabolic genes related to the ECM production of the IVD, including *ACAN*, *COL1*, and *COL2*; (**b**) catabolic genes, including *ADAMTS4*, *MMP3*, and *MMP13*; (**c**) inflammatory genes, including *COX2*, *IL-1β*, and *RANTES*; and (**d**) mechanosensitive genes, including *COMP* and *CILP*. Mean + SD, *n* = 3–5.

**Table 1 materials-16-03173-t001:** Overview of the genes of interest and the primers used for qPCR in this study.

Gene Type	Full Name	Symbol	NCBI Gene ID	Forward and Reverse Primer Sequences
Reference gene	18S ribosomal RNA	*18S*	493779	f—ACG GAC AGG ATT GAC AGA TTG r—CCA GAG TCT CGT TCG TTA TCG
Anabolic markers	Aggrecan	*ACAN*	280985	f—GGC ATC GTG TTC CAT TAC AG r—ACT CGT CCT TGT CTC CAT AG
	Collagen Type 2, Alpha 1 Chain	*COL2*	407142	f—CGG GTG AAC GTG GAG AGA CA r—GTC CAG GGT TGC CAT TGG AG
	Collagen Type 1, Alpha 2 Chain	*COL1*	282188	f—GCC TCG CTC ACC AAC TTC r—AGT AAC CAC TGC TCC ATT CTG
Catabolic markers	ADAM Metallopeptidase with Thrombospondin Type 1 Motif 4	*ADAMTS4*	286806	f—AGA TTT GTG GAG ACT CTG r—ATA ACT GTC AGC AGG TAG
	Matrix Metallopeptidase 13	*MMP13*	281914	f—TCC TGG CTG GCT TCC TCT TC r—CCT CGG ACA AGT CTT CAG AAT CTC
	Matrix Metallopeptidase 3	*MMP3*	281309	f—CTT CCG ATT CTG CTG TTG CTA TG r—ATG GTG TCT TCC TTG TCC CTT G
Mechanosensitive markers	Cartilage Oligomeric Matrix Protein	*COMP*	281088	f—TGC GAC GAC GAC ATA CAC r—ATC TCC TAC ACC ATC ACC ATC
	Cartilage Intermediate Layer Protein	*CILP*	100336614	f—AGG ACT TCG TGC TGT ATG r—CTT GCT CAG GAG GTA GAC
Inflammatory markers	Cyclooxygenase 2	*COX2*	3283880	f—GGT AAT CCT ATA TGC TCT C r—GTA TCT TGA ACA CTG AAT G
	Regulated Upon Activation, Normally T-Expressed, And Presumably Secreted	*RANTES*	327712	f—GTG CGA GAG TAC ATC AAC r—TTA GGA CAA GAG CGA GAA
	Interleukin 1 Beta	*IL-1β*	281251	f—AGT GCC ATC CTT CTG TCA r—CAT TGC CTT CTC CGC TAT T

## Data Availability

Data can be requested from the corresponding authors upon request.

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
