# Peer review of "Repairing Annulus Fibrosus Fissures Using Methacrylated Gellan Gum Combined with Novel Silk"

_materials, 2023, doi:10.3390/ma16083173_

Round 1

Reviewer 1 Report

Tissue section staining for indicated gene (for example, MMP3) would strengthen the study by showing the organization distribution of intentional genes.

Author Response

Dear editors, dear reviewers,

We thank you for your time and valuable and thoughtful comments. We have thoughtfully edited the manuscript and addressed all comments in the revised manuscript and the point-by-point responses below. All changes are tracked and are highlighted in red in the manuscript.

In this rebuttal letter:

  • the comments from the reviewers to the authors are in black.
  • the answers from the authors to the reviewers are in purple.

Reviewer 1:

  1. Tissue section staining for indicated gene (for example, MMP3) would strengthen the study by showing the organization distribution of intentional genes.

Answer: We agree with the reviewer that tissue section staining could further strengthen the study. However, we have to apologize for not being able to provide any histological evaluation for various reasons. Firstly, this study design focused primarily on quantitative biochemical analyses, which we think are of greater weight than semi-quantitative histology, and thus will not significantly increase the soundness of the manuscript. Furthermore, except for significant changes in the IVD’s height, no downstream parameters significantly differed from the control. Therefore, we would also not expect to see any noticeable differences on the protein level. Finally, it would take several weeks or even months to do so as a new set of IVDs would have to be dissected from a fresh bovine tail. The dissected IVDs would then have to be cultured for two weeks and then decalcified for at least one month before the samples could be further processed and sectioned. Since the editors of the journal are already expecting our revised manuscript within 10 days, we fear that this request cannot be granted.

Reviewer 2 Report

1. The major topic of the document can be used to enhance the introduction

2. Citations must be made in the methodology section because the method was taken from the literature; if not, state it in the manuscript.

3. Before doing the study, did the authors weigh the drawbacks of using methacrylated GG?

4. In the line 151, The IVDs were randomly assigned to the control, damage, or repair groups after one day of incubation. How did "random assignment" come about?

5. Figure 2 shows dynamic loading schedule that was repeated every day for 14 days. Compressive pressure is represented by the red line, and torsion by the blue line. Is this a result for a single day or an average over a period of 14 days?

Author Response

Dear editors, dear reviewers,

We thank you for your time and valuable and thoughtful comments. We have thoughtfully edited the manuscript and addressed all comments in the revised manuscript and the point-by-point responses below. All changes are tracked and are highlighted in red in the manuscript.

In this rebuttal letter:

  • the comments from the reviewers to the authors are in black.
  • the answers from the authors to the reviewers are in purple.

Reviewer 2:

  1. The major topic of the document can be used to enhance the introduction.

Answer: Thank you for pointing this out. From our point of view, the major topic of the document was to come up with an approach to repair damaged AF tissue using 2% methacrylated gellan gum (GG) as a filler material and sealing it with an embroidered silk yarn fabric, which resembles the macroscopical structure of natural AF tissue. Therefore, in the introduction, we tried to highlight the need of a novel tissue engineering approach to repair damaged IVDs, since the IVD itself only has a very limited intrinsic healing potential. We then elaborated on the two biomaterials used in this study, namely methacrylated GG and silk. However, as suggested by the reviewer, we have added further content to the introduction on the major topics (methacrylated gellan gum and silk) of this study (page 2, rows 72-89 in the manuscript).

  1. Citations must be made in the methodology section because the method was taken from the literature; if not, state it in the manuscript.

Answer: In addition to the pre-existing citations in the methodology section, we have added the following ones:

  • Chan et al., J Vis Exp, 2012 (page 3, row 101)
  • Frauchiger et al., Eur Spine J, 2018 (page 4, row 166)
  • Xiao et al., Appl Biochem Biotechnol, 2010 (page 5, row 204)
  • Frauchiger et al., Eur Spine J, 2018 (page 5, row 213)
  • Farndale et al., Connect Tissue Res, 1982 (page 6, row 222)
  • Frauchiger et al., J Funct Biomater, 2018 (page 6, row 231)
  • Verdon et al., Anal Biochem, 1995 (page 7, row 237)
  • Reno et al., Biotechniques, 1997 (page 7, row 247)

With these additional citations, all the methods from the literature are now well-supported with adequate references.

  1. Before doing the study, did the authors weigh the drawbacks of using methacrylated GG?

Answer: We thank the reviewer for raising this topic. Before we started with this study, we did some extensive literature research on methalcrylated and plain gellan gum (GG). As stated in the introduction (page 2, rows 78-89) and discussion (page 14, rows 450-464), GG exhibits many properties that are very favourable for tissue engineering, including a great biocompatibility, low production cost, injectability, high hydration, and the option to tune its mechanical properties by means of chemical modifications, like methacrylation for example (Costa et al., Adv Exp Med Biol, 2018 and Stevens et al., Biomater Sci, 2016, Silva-Correia et al., J Tissue Eng Regen Med, 2011). Furthermore, GG has been used to encapsulate and culture human chondrocytes (Oliveira et al., J Biomed Mater Res A, 2010) and methacrylated GG has been suggested by Silva-Correia et al. (J Tissue Eng Regen Med, 2011) and Tsaryk et al. (J Tissue Eng Regen Med, 2017) as a versatile (non-angiogenic, tunable permeability and mechanical properties, etc.) biomaterial to be used in IVD tissue engineering strategies.

We acknowledge that GG-MA, at the proposed formulation, is biomechanically inferior to natural annulus fibrosus (AF) tissue, however, based on all the advantages and positive aspects it brings along as a tunable biomaterial for tissue engineering, we are convinced that the benefits of methacrylated GG outweigh its drawbacks and it can be considered as an encouraging biomaterial for the repair or regeneration of the IVD.

  1. In the line 151, The IVDs were randomly assigned to the control, damage, or repair groups after one day of incubation. How did "random assignment" come about?

Answer: This is a good point. All dissected IVDs from a bovine tail were assigned to one of the groups. To avoid any kind of bias that is related to the IVDs’ size, we made sure that each group was represented by different sized IVDs from different anatomical regions in the tail. Therefore, if for example a relatively big IVD was used as a control sample for one experimental round, we made sure that a smaller IVD was used as a control sample in the next round. However, we realize that this cannot be declared as true randomization. For this reason, we decided to omit this statement in the revised version.

  1. Figure 2 shows dynamic loading schedule that was repeated every day for 14 days. Compressive pressure is represented by the red line, and torsion by the blue line. Is this a result for a single day or an average over a period of 14 days?

Answer: We apologize for any imprecise or missing description. Figure 2 is not a result. It is a representation of what the dynamic loading regime is meant to execute in a single day, which was then repeated for 14 days in total. For this reason, we placed Figure 2 in the “Materials and Methods” section and not in the “Results” section.

To clarify this in the manuscript, we have adapted the caption of Figure 2. The caption now reads as follows:

Representation of the daily dynamic loading regime, which was repeated for 14 days. The red line represents the compressive pressure, and the blue line represents the torsion.

Round 2

Reviewer 2 Report

The authors have made substantial adjustments as suggested. I have no hesitation in accepting this manuscript for publication in the journal.